biochemistry/bioinformatics

manuscripts, charters, parchment, collagen, mass spectrometry

# A biocodicological analysis of the medieval library and archive from Orval Abbey, Belgium

Nicolas Ruffini-Ronzani[1], Jean-François Nieus[1], Silvia Soncin[5], Simon Hickinbotham[5], Marc Dieu[2], Julie Bouhy[3], Catherine Charles[4], Chiara Ruzzier[1], Thomas Falmagne[6], Xavier Hermand[1], Matthew J. Collins[7,8] and Olivier Deparis[3]

[1]Research Centre 'Pratiques médiévales de l'écrit', [2]Mass Spectrometry platform, [3]Department of Physics, and Heritage, Transmissions and Inheritances Institute, and [4]Rare Books Collection, Bibliothèque Universitaire Moretus Platin, University of Namur, 61 rue de Bruxelles, Namur 5000, Belgium
[5]BioArCh, Department of Archaeology, University of York, York YO10 5DD, UK
[6]Rare Books Collection, National Library of Luxembourg, Luxembourg 1855
[7]Department of Archaeology, University of Cambridge, Cambridge CB2 3ER, UK,
[8]GLOBE Institute, University of Copenhagen, Copenhagen 1350, Denmark

MD, 0000-0002-3902-542X; CR, 0000-0001-7958-4528;
MJC, 0000-0003-4226-5501; OD, 0000-0002-2161-7208

**Author for correspondence:**
Olivier Deparis
e-mail: olivier.deparis@unamur.be

Biocodicological analysis of parchments from manuscript books and archives offers unprecedented insight into the materiality of medieval literacy. Using ZooMS for animal species identification, we explored almost the entire library and all the preserved single leaf charters of a single medieval Cistercian monastery (Orval Abbey, Belgium). Systematic non-invasive sampling of parchment collagen was performed on every charter and on the first bifolium from every quire of the 118 codicological units composing the books (1490 samples in total). Within the genuine production of the Orval scriptorium (26 units), a balanced use of calfskin (47.1%) and sheepskin (48.5%) was observed, whereas calfskin was less frequent (24.3%) in externally produced units acquired by the monastery (92 units). Calfskin was preferably used for higher quality manuscripts while sheepskin tends to be the standard choice for 'ordinary' manuscript book production. This finding is consistent with thirteenth-century parchment accounts from Beaulieu Abbey (England) where calfskin supply was more limited and its price higher. Our study reveals that the

making of archival documents does not follow the same pattern as the production of library books. Although the five earliest preserved charters are made of calfskin, from the 1230s onwards, all charters from Orval are written on sheepskin.

## 1. Introduction

The most widespread writing support in medieval Europe was the limed, stretched and scraped skin of calf, sheep and goat. Before the introduction of paper in the fourteenth century, parchment was even the only durable material available for the making of library books and documents. From a zooarchaeological standpoint, a bound codex represents multiple animals drawn from a single livestock generation with a minimum number of individuals equivalent to the largest medieval bone assemblage. Charters, i.e. medieval deeds on single leaves, are usually dated and thus offer exquisite temporal resolution. Although millions of codices and legal deeds written on parchment are still preserved, representing a huge volume of material available for study, they have rarely been considered as primary information on a monastic production, i.e. skins. This is in large part because, until recently, there existed no better method to speciate skins than visual assessment, a reputedly difficult and unreliable process. Catalogues of medieval manuscripts typically describe codices as made of 'parchment' or 'vellum'. Although this latter word (derived from the Old French velin, 'parchment made from calfskin') may occasionally be used to distinguish finer quality skins, 'parchment' and 'vellum' are generally put as synonyms, thus obscuring the fact that medieval scribes were well aware of the precise origin of the parchment they were writing on, with calfskin better valued than the greasier skin of adult sheep. The 1269–1270 accounts from Beaulieu Abbey (Southern England)—a rare piece of direct evidence on skin and parchment procurement and sale—indeed reveal that calfskin commanded a premium price [1].

In recent scholarship, the 'Animal' and 'Material Turns' in the Humanities have focused attention upon the deliberate and informed choices made by parchment-makers and users, who discriminated the young, 'pure' skin from calf against the equivalently sized skins from adult sheep or goat, which evidenced the imperfections of a lived life [2]. More crucially, it is now possible to speciate parchment using zooarchaeological mass spectrometry (ZooMS) [3]. Using these techniques, we have conducted the first-ever comprehensive analysis of a coherent corpus of medieval library books and charters from one single monastic institution, namely the Cistercian abbey of Orval in present-day Belgium. Sixty-eight codices (ninth to seventeenth century though mostly twelfth to thirteenth century) and 59 charters (1173–1359) were analysed, amounting to, respectively, 1490 and 59 sampled folia.

A community of Cistercian monks settled in Orval, in the Belgian Ardennes, in 1131–1132 and still exists today. This small monastery quickly grew in importance and became a major monastic house located at the border between the German Empire and the Kingdom of France. Seventy-four re-bound manuscript books and 59 charters from Orval Abbey are extant today [4,5]. Sixty-eight re-bound volumes in parchment preserved in the National Library of Luxembourg ('BnL' hereafter), which represent the large majority of the collection, and all legal deeds, now in the State Archives in Arlon, were made available for this study and explored through a massive ZooMS analyses campaign between 2017 and 2019. These 68 re-bound volumes contain some 118 identifiable codicological units. We define a codicological unit, abbreviated CU, as a volume, a part of a volume, or a set of volumes whose production may be considered a single operation, prepared in one place, at one time using the same available resources [6,7]. Several clues make the link obvious or possible between these 118 CUs and the medieval library of Orval. First, 26 CUs were copied in Orval. Thirty-three other CUs are of uncertain origin, but their presence in the medieval library is ascertained by ex-libris dated before the middle of the sixteenth century. All the other CUs have a certified provenance at least for the modern period, because they were bound to one of the above-mentioned items, or because their contents are described in a mid-seventeenth-century catalogue, or finally because they appear in miscellanies bearing shelfmarks of the second quarter of the eighteenth century. A list of all 118 CUs under study (with information on date, origin and textual typology) can be found in electronic supplementary material, table S1. While 26 out of these 118 CUs were produced locally by the Orval scriptorium, which was mainly active in the first two-thirds of the thirteenth century, most of them were written elsewhere (in some cases long before the foundation of the monastery), and gradually acquired by the monks through donations, exchanges or purchases.

In the following study, we discriminate between the CUs made in Orval from those created elsewhere. The identification of locally made CUs relies upon Thomas Falmagne's standard catalogue of the Orval

manuscripts [5] (see electronic supplementary material, table S1); three likely attributions to Orval were considered positive. It is worth noting that CUs attributed to Orval are often bound with externally produced CUs within the same volume, which is a common feature of medieval book production. One bifolium from each quire (i.e. a group of bifolia nested together [6,7]) in all 118 CUs was sampled (1490 units), as well as all Orval parchment charters (59 units) (see §2.1). Samples from all but one manuscript were analysed by peptide mass fingerprinting (ZooMS) at BioArCh laboratory, University of York, United Kingdom (see §2.2), while samples from both charter material and the remaining manuscript (ms. BnL 22) were analysed by peptide sequencing at MaSUN platform, University of Namur, Belgium (see §2.3). In order to cope with this considerable amount of samples, an automated classification method (Bacollite [8]) was used for analysing ZooMS data, and the results compared with time-consuming manual analysis. Comparison of the two datasets enabled us to further establish a confidence metric for Bacollite (see §2.4).

Reconstructing the zooarchaeological profile of the entire written patrimony once held by a monastic community offers unprecedented insight into the materiality of medieval literacy and intellectual life. Our exploration of a large set of 'average' library manuscripts and archival records puts the results of earlier works, which were essentially focused on luxury book production, into a broader perspective. This study reveals that the use of calfskin was less prevalent in northwestern Europe than assumed; sheep was far more common than calf. It also sheds light on the tacit and informal rules that guided the choice of animal species by medieval scribes. We argue that the 'quality' level which the commissioner, in this case a Cistercian monastery, expected for its book production may have been a determining criterion. Eventually, our results make clear that from the mid-thirteenth century archival documents did not follow the same pattern as library books, as they were exclusively made of sheepskin.

# 2. Material and methods

## 2.1. Sampling

All sampled codices were classified in the Orval manuscripts catalogue [5] and belong to the collection of the National Library of Luxembourg. All sampled charters belong to the Belgian State Archives in Arlon. Sampling of parchment (manuscript folia or charters) was performed by non-invasive triboelectric extraction of collagen molecules following a previously described method [3]. Briefly, it consisted in gentle rubbing of non-written areas of the parchment surface with a PVC eraser (Mars, Staedler). For each sampling, a new piece of eraser and new nitrile gloves were used, and the table was cleaned with isopropanol in order to avoid any cross-contamination. Eraser crumbs containing collagen, i.e. the parchment sample, were collected in a 1.5 ml Eppendorf tube and stored at 4°C (MaSUN) or room temperature (BioArCh) until its use for ZooMS analysis. For the 68 bound manuscripts, within every codicological unit (118 CUs in total), one sample was taken in each quire of the CU, on the recto of the first folium of the quire. Samples were labelled according to the CU descriptor. The total number of samples amounted to 1490. For the 59 charters, sampling was performed on each charter parchment. The thickness of the parchment was measured using a gauge (0.01 mm accuracy) at six locations on the charter (three points evenly spaced on both left and right edges) and average/ standard deviation values were calculated.

## 2.2. Peptide mass fingerprinting

Samples from all BnL manuscripts except ms. 22 (1490–27 = 1463 samples in total) were analysed at BioArCh laboratory (University of York, UK) by mass fingerprinting. Seventy-five microlitres of 50 mM ammonium bicarbonate buffer ($NH_4HCO_3$, AmBic, pH 8) and 1 µl of trypsin (0.4 µg µl$^{-1}$) were added to the eraser crumb samples and incubated at 37°C for 4 h to perform collagen digestion. The Eppendorf tubes containing the samples were then spun down on a benchtop centrifuge at maximum speed and the trypsin action was stopped by adding 1 µl of 5% vol/vol trifluoroacetic acid (TFA). The supernatant were transferred into new Eppendorf tubes and C18 resin ZipTip® pipette tips (Millipore) were used to extract the collagen peptides, which were then eluted into 50 µl of 50% acetonitrile (ACN) and 0.1% (v/v) TFA (conditioning solution). One microlitre of eluted peptides was spotted in triplicates on a ground steel plate and mixed with 1 µl of matrix solution (α-cyano-hydroxycinnamic acid), together with a calibration mixture [3,9]. The MALDI-ToF analysis was performed using a

Bruker Ultraflex III mass spectrometer at the Centre of Excellence in Mass Spectrometry facility at the University of York.

## 2.3. Peptide sequencing

Samples from all charters (59) as well as those from ms. 22 (27 samples) were analysed by peptide sequencing at MaSUN mass spectrometry platform (University of Namur, Belgium). Collagen was extracted and digested by adding 50 µl of $NH_4HCO_3$ 50 mM buffer to the sample and 200 ng of trypsin (Promega). The samples were incubated for 4 h under light agitation. The samples were acidified with a solution of TFA to a final concentration of 1% (v/v). Supernatant containing the peptides were transferred to a ZipTip® C18 (Millipore). After washing and conditioning of the ZipTip® according to the manufacturer's instructions, the peptides were loaded and desalted with a solution of $H_2O$, 0.1% TFA (v/v). Peptides elution was done with 10 µl of 80% ACN/0.1% TFA (v/v). All samples were then vacuum dried (Heto) and recovered with a solution of 2% ACN/0.1% TFA (v/v). All samples were analysed using liquid chromatography (UltiMate 3000, Thermo Fisher) coupled to electrospray tandem mass spectrometry (MaXis Impact UHR-TOF, Bruker) (LC-MSMS). The digests were separated by reverse-phase liquid chromatography using a 75 µm × 150 mm reverse-phase column (Acclaim PepMap 100 C18). A 20 min gradient was used to separate the peptides. The column effluent was connected to a Captive Spray (Bruker). In survey scans, MS spectra were acquired for 0.5 s in the $m/z$ range between 50 and 2200. The 10 most intense peptides $2^+$ or $3^+$ ions were sequenced. The collision-induced dissociation (CID) energy was automatically set according to mass-to-charge ($m/z$) ratio and charge state of the precursor ion. MaXis and Thermo Systems instruments were piloted by Compass HyStar 3.2 (Bruker). Peak lists were created using DataAnalysis 4.0 (Bruker) and saved as an mgf file for use with Mascot 2.4 as search engine (Matrix Science). Enzyme specificity was set to trypsin, and the maximum number of missed cleavages per peptide was set at one. Hydroxylation (KP) and oxidation (M) were allowed as variable modification. Mass tolerance for monoisotopic peptide window was 7 ppm and MS/MS tolerance window was set to 0.05 Da. The peak lists were searched against a home-made collagen protein database and a contamination protein database. Scaffold software (Proteome Software) was used to validate protein and peptide identifications, and also to perform the search of species marker peptides. Our species marker database contained specific peptides that allowed us to differentiate *Capra hircus* (goat), *Ovis aries* (sheep) and *Bos taurus* (calf).

## 2.4. Species identifications

Manual species identification was performed by screening the mass spectra for peptide $m/z$ markers. To do so, we used the open-source software mMass (www.mmass.org) [10], setting the signal-to-noise threshold at 3.0 and the relative intensity threshold at 0.3. The $m/z$ markers correspond to the peptides isolated from collagen $\alpha_1$(I) and $\alpha_2$(I) following the protocol described in the previous section. Cattle (*Bos taurus*), i.e. calf here, can be distinguished from other mammals by the peptide of the $\alpha_2$(I) chain at $m/z$ 1192.6 + 1208.6 and that of the $\alpha_2$(I) chain at $m/z$ 2853.5 + 2869.5. The same two peptides in sheep (*Ovis* sp.) and goat (*Capra* sp.) collagens have $m/z$ 1180.6 + 1196.6 and $m/z$ 2883.5 + 2899.5, respectively [11]. Sheep and goat can be further discriminated by the peptide of the $\alpha_2$(I) chain with $m/z$ 3017.4 + 3033.4 for sheep, and $m/z$ 3077.4 + 3093.4 for goat [12]. When this last peptide was not present, the sample was identified as mixed 'sheep/goat'. In the case of absence of any relevant peptide, the sample was categorized as 'undetermined'.

The manual interpretation of ZooMs results for such a high number of parchment samples gave us the opportunity to use a new, automatic method of species identification, called Bacollite [8] to validate the manual analysis. This automated approach has two stages. The first one combines a cross-correlation of observed MS spectra with the theoretical distribution of a set of peptides that are used for discrimination. The second one involves an iterative thresholding scheme which compares the correlation values and yields a score for each candidate species that can be used for classification. For the current study, the entire dataset was processed using the same set of discriminatory peptides that were used in the manual analysis described above.

With the manual classification to hand, our approach to species ID validation was as follows. Firstly, the automated classification was compared with the manual classification and the dataset was divided into those samples where the manual and automated interpretations agreed and those where they disagreed. This allowed us to obtain a distribution of scores for these two classes, as shown in

electronic supplementary material, figure S1. The majority of manual and automated classifications agreed, and after correcting a small number of recording errors in the manual data, it was clear that samples with disagreement on classification all had a low Bacollite score. Accordingly, a threshold score of 10 was agreed, below which the manual interpretation was accepted as the final decision on species ID. The small number of samples (20) above this threshold that had no manual ID were re-examined, and where appropriate the Bacollite classification was accepted. The distribution of Bacollite scores in each CU is shown in electronic supplementary material, figure S2. The plot shows that calf tends to score more highly than sheep or goat. This is because calf have fewer peptides in common with sheep or goat, and the scoring scheme is designed to reflect the 'discriminatory power' of the sample. The distribution of scores seems highly variable among the CUs, however.

## 2.5. Statistics

Results of species identifications were classified into six categories: three were associated with pure species ('calf', 'sheep', 'goat') and the other ones were related to ambiguous results ('sheep/goat', 'undetermined', 'not analysed'). A first series of statistical analyses was carried out at the quire level. Percentages (number of samples falling into a given category divided by the total number of samples considered) were calculated while sorting on manuscript production place (Orval scriptorium or outside). A second series of statistical analyses was carried out at the CU level. For a given CU, the numbers of samples falling into each species ID category were calculated. The category with the maximum number of samples was regarded as the dominant category for the CU. A percentage of occurrences of the dominant category ($p_{max}$) was then calculated. Based on this percentage, two descriptors were assigned to the CU: a CU type and a dominant species. If $p_{max} > 85\%$ and the category was associated with a pure species, then both CU type and dominant species descriptors inherited the name of the species. Otherwise ($p_{max} \leq 85\%$ or the category was related to ambiguous results), different cases were considered. If the dominant category was neither 'undetermined' nor 'not analysed', the CU type descriptor was assigned to 'mixed', otherwise it was assigned to 'undetermined'. Furthermore, as it could happen that the maximum number of samples fell into a non-pure species category, the dominant species descriptor inherited the name of that category as far as $i_{max} > 85\%$. On the other hand, if $p_{max} \leq 85\%$, the dominant species descriptor was assigned to 'none'.

## 2.6. Codicology

In order to refine our statistical analysis, we have defined three main additional features for CUs. The first feature, called 'thickness' herein (called 'consistance' by Muzerelle & Ornato [13]), is based on the number of folia a given CU contains. According to [13], most of the medieval manuscripts counted between 100 and 200 folia. CUs of more than 200 folia only represented a small part of the Orval corpus (between 10% and 20% according to the period of production) as did CUs of fewer than 100 folia (between 15% and 25%). CUs were regarded as 'very thin' when they counted less than 10 folia. Short and isolated texts were usually written on these CUs. They were regarded as 'thin' when the CUs counted between 11 and 100 folia, 'medium' when they counted between 101 and 200 folia, and 'thick' when they counted more than 200 folia. The second feature, called 'quality' herein, was based on layout, scribe's skill (calligraphy) and decoration. Regarding the layout, we considered that the best layouts contributed to the readability of the texts. The skill, or even the zeal, with which the scribes wrote the texts was a second important aspect. Indeed, some writings were copied more carefully than others. Finally, the decoration of the manuscripts matters. The illuminated manuscripts were regarded as 'better' than the manuscripts without decoration or containing only coloured initials. We have attributed notes to each CU for each of these criteria and, then, brought the notes to scores ranging from 0 to 10. Based on this score, we have set up five categories of manuscripts, i.e. 'very low quality' (score lower than 3), 'low quality' (score higher than or equal to 3 and lower than 5), 'medium quality' (score higher than or equal to 5 and lower than 6.5), 'good quality' (score higher than or equal to 6.5 and lower than 8) and 'superior quality' (score higher than 8). The third feature was a typology based on the manuscript topic: eight types of texts were defined (Bible, liturgy, grammar and rhetoric, sciences, narrative texts, law, preaching, theology) plus a category 'Other', which was composed of normative texts and letter collections.

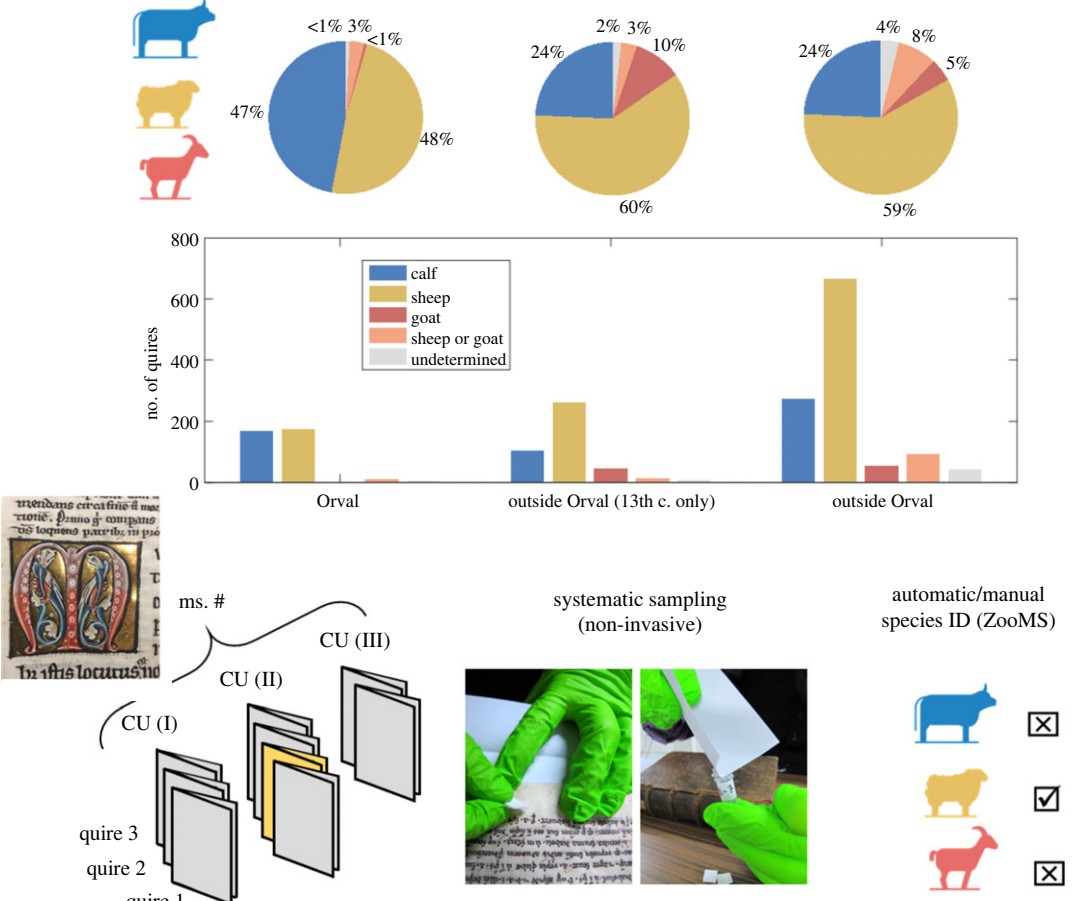

**Figure 1.** Distribution of parchment animal skins among the 68 manuscripts of the Orval Abbey library. Proportions (top chart) and counts (bottom chart) in each species ID category (calf, sheep, goat, sheep or goat, undetermined) were calculated at the quire (sample) level and are displayed according to the production place and the production period of codicological units (CUs). The illustration below the charts depicts the systematic non-invasive sampling of CUs at the quire level. Collagen samples (1490 in total) were taken by gently rubbing parchment on the recto of the first folio of each quire composing the CUs and then analysed by ZooMS.

# 3. Results

## 3.1. Species variation in and outside the Orval scriptorium

The manuscripts from Orval library are made of calfskin, sheepskin and goatskin, the most commonly used species in the medieval West. In total 1490 bifolia were analysed by ZooMS (electronic supplementary material, table S2) and statistics were performed on species ID results (see §2.5). Approximately 56.4% are sheep, 29.7% calf, 3.8% goat, 7.0% sheep/goat (i.e. the ZooMS spectra were too poor for assignment), and 3.1% are left undetermined or were not analysed. However, significant differences are observed between the 26 CUs which were identified by Falmagne as Orval production, and the 92 CUs which were not (figure 1). The Orval scriptorium seems to have made more frequent use of calfskin than is evident in the texts acquired by the library. Almost half (47.1%) of the production of the scriptorium at Orval (mainly active in the early to mid thirteenth century) was on calfskin. This contrasts with only a quarter (24.3%) of the folium from acquisitions made in the thirteenth century, a similar fraction as observed from all imported texts. Consequently, sheepskin appears less predominant in Orval-made CUs (48.5%) than in non-Orval CUs (58.9% of thirteenth-century CUs, 60.4% of medieval CUs). In any instance, these data do not corroborate the trend argued for manuscripts from Benedictine monasteries in the Scheldt valley by Steven Vanderputten in 2005 (exclusive use of sheep during the tenth and eleventh centuries, and dramatic shift to calf in the first half of the twelfth century), inferring from naked eye inspection of 42 codices dedicated to lives of

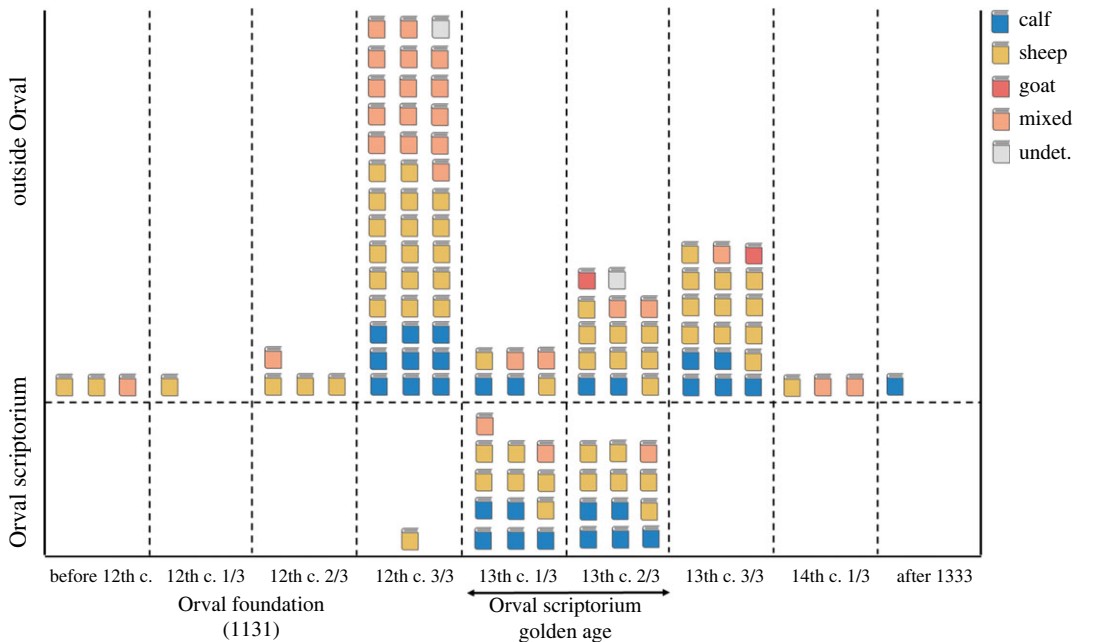

**Figure 2.** Timeline representation of animal skins among the codicological units produced by the Orval scriptorium and outside. For each codicological unit, a CU type descriptor (calf, sheep, goat, mixed, undetermined) was assigned from the percentage of occurrence of the dominant category, which was calculated from species ID data obtained at the quire level.

saints [14]. Eventually, the Orval monks hardly ever made use of goatskin (0.6%), a type of parchment which also remains uncommon in non-Orval productions (10.2% of thirteenth-century CUs, 4.8% of medieval CUs).

The above-mentioned statistics were performed at quire level. Yet, all following analyses will be made at CU level, with 'mixed' CUs treated separately, and labelled as such when their prevailing species falls below a given percentage (see below). Individual species analyses at CU level are listed in electronic supplementary material, table S3. The broader use of calfskin in Orval Abbey is more evident when data are plotted on a timeline (figure 2), and organized by thirds of a century (early, mid and late) according to the approximate date of each CU (see electronic supplementary material, table S1). The use of calfskin at Orval remains constant throughout the 'golden age' of the local scriptorium (five CUs in the first third of the thirteenth century, five CUs in the second third), which, by contrast, coincides with a period when very few non-Orval CUs were made of calf. The Orval scriptorium embraced sheepskin too, for 12 CUs were written on this type of parchment in the first two-thirds of the thirteenth century. Yet, compared with imported texts, Orval selected calfskin over sheepskin. The only pair of CUs essentially made of goatskin (CU 30-IV: 100% goat; CU 74: 97.62% goat; see electronic supplementary material, table S3) did not emanate from the Orval scriptorium, whose scribes only used goatskin once, into a mere two quires of a single CU (CU 35-2, CU 35-3; see electronic supplementary material, table S3). In England, Dyer has documented the decline in goat husbandry during the medieval period, in part due to their increasing legal liability [15], while Salvagno & Albarella use zooarchaeological evidence to also conclude that goats were rarely raised in the same period, but that their skins were being imported from the continent [16].

Our timeline (figure 2) also accounts for the existence of some 27 CUs which are composed of several skin types. For instance, CU 35, written in Orval in the first third of the thirteenth century, is made of both calfskin (four quires, and one final leaf) and goatskin (two quires), while CU 72-II, created elsewhere in the final decades of the twelfth century, has sheepskin (seven quires), calfskin (four quires) and some goatskin (one quire). Such heterogeneity may have been a fairly common feature of medieval book production, and is seen in a Glossed Luke produced in Canterbury during this period [17]. In what follows, CUs with fewer than 85% of their quires made of one skin type are considered 'mixed' CUs (see §2.5). It must therefore be noted that, when a CU is said to be 'homogeneous', it essentially means that one skin type prevails among the quires of this CU (greater than 85%); it does not rule out the presence of some quires made of another skin type (less than 15%). By making the balance between mixed and homogeneous CUs, we can assess the overall homogeneity of the corpus.

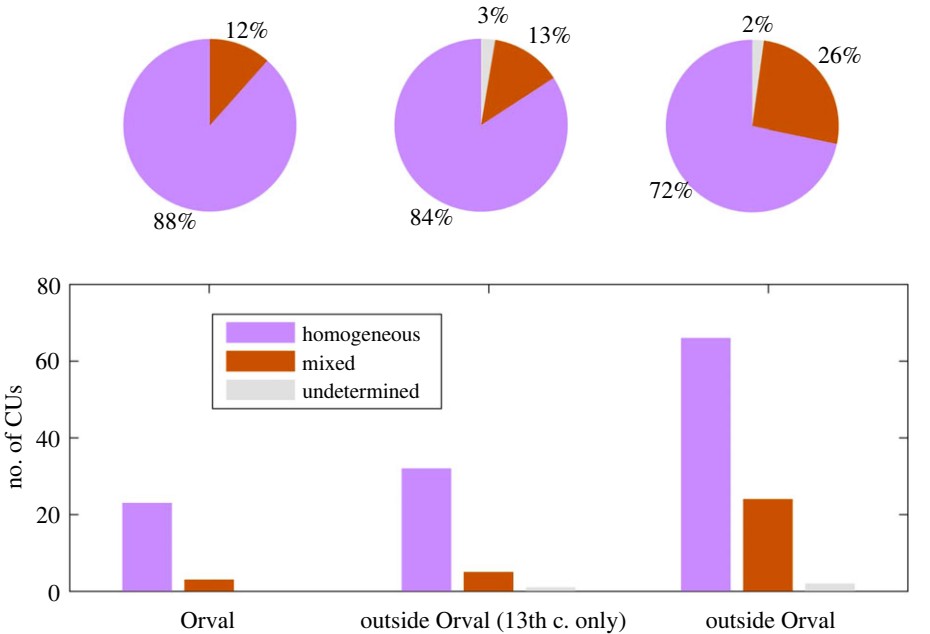

**Figure 3.** Homogeneity of codicological units from the Orval Abbey library. Homogeneous CUs refer to codicological units whose CU type descriptor falls into calf, sheep or goat categories.

As shown in figure 3, CUs produced in Orval are slightly more homogeneous than non-Orval CUs (88.5% versus 71.7%). Popular combinations of species within mixed CUs are sheepskin and calfskin (6/27 = 22.2%), or a mix of the three common skin types (5/27 = 18.5%). Calfskin and goatskin are only combined once. Sheepskin and goatskin may possibly be too, but, due to uncertain species ID (see Material and methods), we had to consider 'mixed' 13 CUs (13/27 = 48.1%) made from quires assigned to sheep, goat and 'sheep/goat'. In a further pair of mixed CUs, undetermined species ID prevailed.

In his catalogue, Falmagne was able to determine a plausible geographical origin (place or area of production) for nearly half of the manuscripts once kept in Orval library (47 CUs, see electronic supplementary material, figure S3). Twenty-six CUs were made in Orval, 13 in Northern France (mostly Champagne), four in Paris, three in medieval 'Lotharingia' (approximately today's Benelux, Eastern France and Rhineland), and one in Italy. Correlative mapping of skin types confirms that calf was used more widely in Orval than in other represented areas (except in Paris, albeit a small dataset). Calfskin even appears either absent or rare in CUs produced in regions close to Orval (Lotharingia, Northern France), which may have seemed more inclined to use sheepskin.

## 3.2. Possible motives for skin choice in book production

In order to address the question as to what factors guided the choice of a given type of parchment for manuscript book production in the twelfth to thirteenth centuries, we have explored the possible correlations between skin type and several design features of the CUs from Orval library. Some criteria were found less relevant than expected. For instance, there is no clear correlation with the format (dimensions) of CUs (electronic supplementary material, figure S4), apart perhaps from a slight preference for calfskin in the making of small-sized manuscripts (7/22 = 31.8%). Sheepskin represents between a third and a half of the CUs in each of the first three format categories defined in Bozzolo & Ornato's typology (10/22 = 45.5% in small, 38/69 = 55.1% in medium–small, and 10/25 = 40.0% in medium–large CUs) [18].

A more significant feature may be the thickness of CUs, which depends on the number of folia in a given CU [13]. On this basis, we have established an *ad hoc* scale of four CU thickness levels (see §2.6). It is worth noting that a majority of CUs from Orval library do not fit into Muzerelle & Ornato's standard model [13], for they have fewer than 100 folia (16.1% are in category 1: 'very thin', 39.8% in category 2: 'thin', and only 33.0% belong to category 3: 'medium'). Beyond this, three observations are of particular interest (electronic supplementary material, figure S5). First, more than three-quarters of 'very thin' CUs

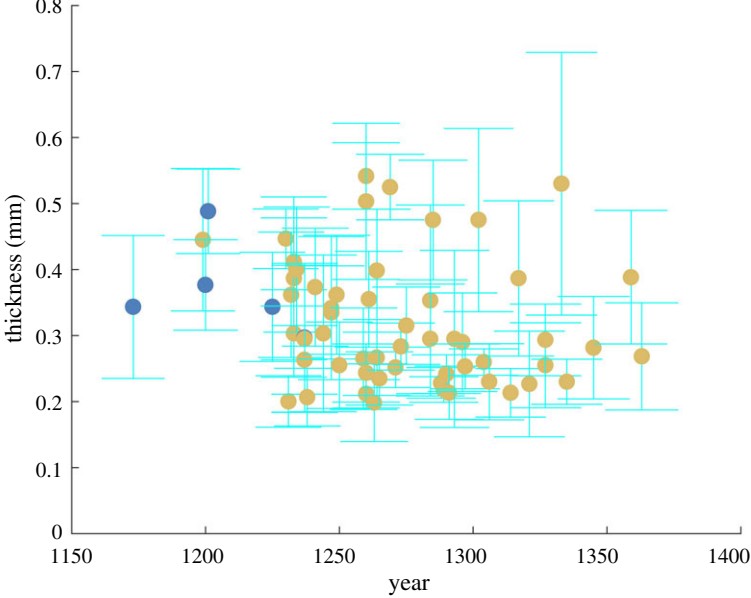

**Figure 4.** Timeline of parchment thickness and animal skins used in the charters from the Orval Abbey. Animal skins were identified by ZooMS (blue dots: calf, yellow dots: sheep). Errors bars are standard deviations calculated from thickness measurements at six different points on the charters. One charter (c46, see electronic supplementary material, Table S4) has no date though it was produced in the thirteenth century (for plotting this timeline, its date was arbitrarily set to 1250).

were made in sheepskin (78.9%). Second, the rate of calfskin is roughly constant in the next three categories (respectively 29.8%, 28.2% and 23.1% in 'thin', 'medium' and 'thick' consistency CUs). Third, mixed CUs are proportionally over-represented among 'medium' CUs (41.0%).

We have also gauged the potential link between skin choice and textual content (electronic supplementary material, figure S6). To this end, a simplified typology of nine text categories has been set up, including a residual category 'Other' (see §2.6). Despite the limited size of the Orval corpus, which does not allow definitive conclusions here, some trends are clearly emerging. In the first instance, all biblical manuscripts—the most sacred books thus—were written on calfskin. Conversely, sheepskin was commonly used for grammar and rhetoric treatises, and also praised for law (8/16 = 50.0%), scientific (3/6 = 50.0%), and narrative manuscripts (10/15 = 66.7%). In the intermediate branches of theology, liturgy and preaching, calfskin, sheepskin, as well as mixed CUs occur in approximately identical proportions.

Ultimately, though, the most determining factor in the choice of skin type probably was the overall quality of the manuscript in the making. We are well aware that, in the eyes of experts in medieval manuscript books, objectifying 'quality' is an extremely delicate issue. What we have done here is to assess quality on three key criteria: layout, calligraphy and decoration [19]. These criteria were subjectively ranked on a scale ranging from 1 to 10, then converted into five quality levels : 'very low quality', 'low quality', 'medium quality', 'good quality' and 'very good quality' (see §2.6). The result appears especially meaningful, as it reveals a strong correlation with skin type (electronic supplementary material, figure S7). Indeed, a majority of 'very good quality' CUs were written on calfskin (7/11 = 63.6%), while a similar proportion of 'very low' and 'low quality' CUs (5/8 = 62.5% and 13/22 = 59.1%) were produced on sheepskin.

## 3.3. Charter production in and outside the Orval 'chancery'

The 59 parchment charters left from the charter collection of Orval Abbey were also analysed. These legal deeds were received by the monks between 1173 and 1363; most of them (43 units) date from the thirteenth century [4]. They were authored by various persons and institutions, mostly from adjacent areas, such as local lords, city councils or bishops, but there are also seven papal bulls among them. Therefore, in terms of geographical origins, this set of archival documents represents a heterogeneous corpus. Since the writing of charters in the Orval 'chancery' has not yet been studied, we do not know exactly which charters were manufactured within the monastery, and which were produced

outside, by their author's employees. Indeed, in the Middle Ages, charters were frequently prepared by their beneficiaries, especially when their nominal author was not prominent enough as to have a writing office or personnel of his own [20]. While papal bulls were certainly produced by the pontifical chancery, most deeds issued in the name of second-rank aristocrats, such as Thierry of Walcourt in 1200, were plausibly made by the Orval monks.

Peptide sequencing revealed that 54 charters are made of sheepskin, while a mere five are in calfskin (figure 4). Goatskin is completely absent from the Orval charters. The five charters in calfskin all predate the mid-thirteenth century: they were issued in 1173, 1200, 1201, 1225 and 1237 (electronic supplementary material, table S4). Their authors were lay and religious dignitaries from Lotharingia, the political entity where Orval Abbey was settled. Remarkably enough, from the 1230s onwards, all charters granted to Orval, including papal bulls, were made of sheepskin. This monopoly of sheepskin is congruent with the results of previous research on English late medieval and early modern legal deeds and French fourteenth-century account books, which were all written on sheepskin [3,21]. It seems that in the Carolingian age (eighth to tenth centuries), sheep, calf and goat were all regarded as suitable for charter-making [22,23]. Things thus changed at some point. The Orval evidence suggests that this happened in the early decades of the thirteenth century, although it entails few charters from this very period.

The thickness of the Orval charters was also examined (see §2.1) [24,25]. Systematic measurements (figure 4) did not reveal any significant difference between charters in sheepskin and charters in calfskin. What they taught us, however, is that the thickness of charters starts decreasing from the beginning of the thirteenth century. This observation is in accordance with the measurements taken from a substantial sample of about 400 twelfth- and thirteenth-century papal bulls by Frank M. Bischoff in 1993 [23]. A decrease in thickness might therefore have been a general trend in Europe during the thirteenth century.

## 4. Discussion

### 4.1. A first glimpse into 'ordinary' medieval book production

Our complete investigation of the Orval library is a first, important step towards a global understanding of parchment use in twelfth- and thirteenth-century 'ordinary' medieval book production in northwestern Europe. In addition to offering an insight into a wide variety of codices, the Orval corpus presents a strong coherence from a geographical standpoint, as most of its components were created in the same area (Northern France and ancient Lotharingia, except one imported Italian manuscript). The conclusions of the present study logically differ from those reached by the first, ground-breaking survey of parchment manuscripts by ZooMS, published in 2015 by researchers from the University of York, UK, and primarily focused on thirteenth-century 'pocket' Bibles (i.e. a select category of extremely small-sized books) originating in France, England and Italy [3]. Based on the less exclusive and more clustered Orval evidence, it is now clear that sheepskin, and not calfskin as suggested by pocket Bibles, was the most commonly used skin type in book production during the central Middle Ages. This was at least the case between the Seine and Rhine rivers, yet probably also well beyond. Our results further substantiate the common view that goatskin parchment was scarcely used in the northern part of Europe. In this respect, however, a discrepancy can be observed between the 26 CUs recognized by Falmagne as products from the Orval scriptorium during its 'golden age' (ca 1200 to 1260), and CUs originating from other (mostly unidentified) writing places. Indeed, while most of the 20 non-Orval CUs dating from the same period were written on sheepskin, the ratio of sheepskin to calfskin is almost equal to unity within Orval-made CUs. Whatever the exact reasons for this situation (see below), it suggests that ZooMS techniques, when applied to a large corpus, may help to unveil discriminating features for particular book productions.

The prominence of sheep- and calfskin in Orval-made books is congruent with cattle breeding practices in the Ardennes region in the high and late Middle Ages. It is interesting to compare the size of this study from a zooarchaeological perspective. In a synthetic study of 105 sites excavated in London over the space of two decades, Thomas and co-workers were able to analyse measurements of a total of 4300 individual cattle bones and horn cores and 1951 ovicaprid bones (sheep and sheep goat) [26]. Of these 129 cattle and 97 ovicaprid dated to the earliest time period (1220–1300), about the length of the Orval golden period. Like several horn cores and bones may belong to one single animal, multiple folia may be produced from one single skin. The numbers in the 20 non-Orval CUs (158 acquired folia out of 1373 in total) dwarfs typical

zooarchaeological studies. Here, unlike for local production, four times as many were derived from skins of sheep (118) as calf (29), with a further three goatskins. From a zooarchaeological standpoint, young calf and goat bones are rarely reported from archaeological sites in northwestern Europe in this period [27] and the 4 : 1 ratio of sheep to cattle is higher than is typical for similar synthetic surveys of the period [28] where the bone proportions of sheep and cattle in England are roughly equal. Available evidence from Orval Abbey does not allow historians to characterize its livestock in absolute or relative terms, except for pigs (at least 400–500 units in the second half of the thirteenth century [29]). All we know is that the Cistercian monks of Orval managed their presumably numerous livestock with great care, jealously defended their privileges in the nearby forests, and were always keen to gain access to new pastures [29]. Closer estimates are only possible for a few religious houses in the same region, such as the Dominican priory of Marienthal, some 50 kilometres west of Orval [30]. According to surviving documents, flocks of goats were rare and small-sized, which evidently explains why goatskin was never used in the Orval scriptorium, except for a couple of isolated, erratic quires [31]. Bovine herds and ovine flocks were both present in the Ardennes, but not in similar proportions, as sheep flocks were by far larger than bovine herds. For instance, in 1321, Marienthal owned 597 sheep, including 132 lambs at least, against 74 adult bovines and 45 calves.[1] Easier access to sheepskin may therefore account for its overall predominance in book production (as well as, to some extent, for its ever more exclusive use for charters: see below). Moreover, according to the aforementioned thirteenth-century financial records from Beaulieu Abbey (England), sheep parchment was cheaper than calf (and goat) parchment, arguably because it was considered lower quality [1]. A comparison can be made with the meat market in medieval France, as calf meat was usually more valued than sheep meat [32]. In this context, the special preference for calf parchment demonstrated by Orval monks in the thirteenth century cannot have originated from an economic reasoning. An explanation for this is to be found elsewhere.

## 4.2. Calf or sheep? Quality and correlated factors

Among the above-discussed factors which possibly influenced the scribe's choice for a particular type of skin, none was decisive in itself. Rather, beside merely practical issues such as availability and cost effectiveness, which may of course have been determining in some instances (though probably not so much in a wealthy and resourceful Cistercian monastery), this decision is likely to have resulted from an intricate combination of factors, each driven by its own logics. We have seen that a certain sense of quality must have played a key role. Basically, our survey has shown that the better a manuscript was executed, the stronger the likelihood that it was written on calfskin, which corroborates an existing conventional wisdom among codicologists [22]. As an extreme example, the two most valued manuscripts in our corpus, which were also the most lavishly decorated (CUs 100-I and 138, with gold used for the illuminations), were both made of calfskin. Conversely, average to low-range books had a good chance of being written on sheepskin. A notion of prestige must therefore have been attached to calf parchment. This is reflected in the Orval corpus by the correlation between animal species and textual content. All biblical manuscripts, that is by far the most revered books in a monastic library, are made of calfskin (as were most thirteenth-century pocket Bibles, though not necessarily for identical reasons), while sheepskin was preferentially used for non-religious texts regarded as 'work tools' of lesser status, such as grammar and rhetoric treatises, as well as law, scientific and narrative manuscripts.

But what did make calfskin more suitable than sheepskin for high-quality productions? There is no clear-cut answer to this question. The thickness of resulting parchment was not necessarily at stake: calfskin is naturally much thicker than sheepskin [33], yet medieval parchment-makers were equally able to make very thin sheets from both species [3]. Our observations on the thickness of CUs are difficult to interpret in this respect. There is evidence that medieval scribes were deeply concerned to make books that were easy to handle and looked good [13]. Small-sized manuscripts must not be too thick, otherwise they would have been poorly adapted to consultation and have looked disgraceful (tower-books), whereas large-sized manuscripts had to be thick enough as not to resemble 'strip-books'. But the reason why small manuscripts in the Orval library are proportionally more often made of calf, while nearly all 'thin' CUs were written on sheepskin, remains unclear. It is well possible that calfskin first and foremost appeared as a better raw material for parchment-making because it has less fat than sheepskin, and was therefore a better surface for ink and paint [22,34]. One might also think of the

---

[1]Here is the complete counting in Marienthal, according to Yante [30]. On 871 animals (100%): 29 stallions (3.5%), 17 foals (2%), 74 cows, bulls and oxen (8.5%), 45 calves (5%), 30 goats (3.5%), 79 pigs (9%) and 597 sheep, among which 132 lambs at least (68.5%).

visual aspect of neatly prepared parchment from calf, which enjoys the reputation of being especially spotless, white and smooth [2]. Further research is needed along these lines.

## 4.3. Parchment charters: a reality apart

Crucially, our study reveals that the making of archival documents did not follow the same pattern as the production of library books. While calf-, sheep- and even goatskins were used concomitantly in book production during the central Middle Ages, the charters collected by Orval monks were exclusively made of sheepskin from the 1230s onwards. Charters made of calf still occur at the turn of the twelfth and thirteenth centuries, but then disappear. This trend is all but specific to the local 'chancery', for a substantial (though not quantifiable) part of the deeds kept in Orval, including several papal bulls, were written in other places. Analyses carried out on two fourteenth-century sets of French and English documents have shown that in this period, other kinds of archival records, such as account books and rolls, were also made of sheepskin [3,21]. Yet, earlier studies (based on visual assessment) of charter material from St Gall Abbey (now Switzerland) and the region of Lucca (Tuscany, Italy) pointed out that, in the first part of the medieval era, calf and goat were still commonly used beside sheep [22,23,35]. The Orval evidence may therefore bear witness to a major shift in the early decades of the thirteenth century—to be confirmed by further research on other charter collections—towards a growing monopoly of sheepskin in the field of legal deeds and other administrative documents. There were exceptions, however. The papal chancery in Rome, a very conservative institution, reportedly continued to use goat parchment in the thirteenth century (80–75% sheep, 20–25% goat, according to Bischoff [23]). Case studies indicate that north of the Alps, some types of solemn deeds, such as imperial diplomas or city privileges, were still made of calfskin [36,37]. This is the case, for instance, of the Austrian *Privilegium maius* and its accompanying set of forged diplomas (1359–1360). The forgers went as far as trying to simulate goat parchment by reworking a calfskin, since they knew that their supposed eleventh-century diploma should have been issued in Italy, and consequently written on goat parchment [37]. This is a spectacular reminder that medieval scribes were acutely aware of animal species, and able to discriminate between them [2,38].

The growing use of sheepskin in archival production coincides with the 'revolution of writing': from the late twelfth century onwards, written records dramatically proliferated in the medieval West [39]. In this context, the need for parchment became more and more acute, which may help explain the success of cheaper sheepskin against calfskin. In addition, everyday, ephemeral administrative writings did not require premium quality support. Yet, there may be more compelling reasons for the near-eradication of calf parchment from pragmatic literacy. According to a celebrated late twelfth-century English treaty (*Dialogue of the Exchequer*), sheep parchment was seen as an advisable protection against forgery, thanks to its propensity to delaminate when part of the supported text was being erased for rewriting [40]. This feature was very convenient indeed in the domain of legal deeds. There might also be a link with the introduction and widespread raising of fattier wool sheep.

The moderate downward trend in thickness observed among thirteenth-century charters might reflect a tendency towards an improved preparation of the parchment [22]. Yet, the crucial information about thickness is that charters are significantly thicker than book leaves (although they only needed to be prepared for writing on one side), possibly because, as reported by Conrad of Mure in the late thirteenth century [41], they were expected to be slightly rigid. Parchment sheets aimed at the writing of documents were therefore made, prepared or selected in a different way. Charter parchment is definitely a reality apart.

## 5. Conclusion

Systematic ZooMS analysis of a large and coherent corpus of manuscripts and charters from one single monastic institution was carried out on a very high number of samples (1490). This large amount of samples set an unprecedented challenge to species identification from manual analysis of ZooMS data. Beside manual classification, an automated analysis method was used to consolidate species ID results. There are two key advantages to the automated approach. Firstly, it has a much smaller time cost and training overhead. Secondly, it yields for each sample a score which can be interpreted as a confidence metric in the validity of the classification. This study demonstrates that the scoring system is a useful metric for confidence in the automated identification of species for parchment data, and allows expertise to be targeted at samples which are more difficult to interpret.

This first-ever complete survey of a monastic library and archive is also a major breakthrough in our understanding of parchment use in medieval Europe. It makes the case that calf was not the most commonly used animal species for parchment-making in continental northwestern Europe in the central Middle Ages, and confirms that goatskin was virtually absent north of the Alps. It is rather the skin of sheep that appears as the standard raw material for 'ordinary' manuscript book production. Remarkably, sheep parchment was even more prevalent in the making of archival documents: from the 1230s onwards, all surviving charters from Orval Abbey were written on this support. A complex combination of factors may account for the success of sheepskin. Sheep flocks were far more numerous in northwestern Europe than bovine herds and goat flocks, which most probably made sheep parchment more available and less expensive, and therefore best suited for medium- and low-range book production. A booming need for writing material in the sphere of administration after 1200 (revolution of writing) certainly reinforced the attractiveness of cheaper— and forgery-proof?—sheep parchment. Calf parchment, which was presumably praised by medieval scribes as an upper quality material, may have been reserved for the making of valuable manuscripts.

Indeed, within the Orval library, the more select a manuscript appears, the greater the chance that it is written on calfskin. This is probably one of the most striking underlying principles which medieval book production obeyed, in terms of species selection. As a corollary, a relation can be observed between the textual content of a given book and the type of parchment it is made of. The most valued and longer works, to begin with biblical texts, were made of calfskin, whereas secular writings regarded as mere 'work tools', such as grammar books, were generally written on sheepskin.

Non-invasive analytical techniques such as ZooMS open new fields in the study of medieval written production. Combined with codicology and diplomatics, these new techniques allow to reinvestigate the production of manuscript books and charters. In addition, they also contribute to improving our understanding of crafting techniques and cattle breeding in the Middle Ages. It is necessary now to analyse new coherent corpora of manuscripts and charters from all over Europe, as we have done for Orval Abbey. By multiplying the case studies, it will be possible to draw a map of parchment production and use in medieval Europe.

Data accessibility. The datasets supporting this article have been uploaded as part of the electronic supplementary material.

Authors' contributions. X.H., J.-F.N., T.F., M.J.C., O.D. conceived the study; T.F. realized the codicological study; C.C. organized parchment sampling; S.S., M.D., J.B. performed ZooMS analysis; S.S. (S.H.) conducted manual (automated) species identification; O.D. wrote dedicated scripts for statistical analysis; N.R.-R., J.-F.N., C.R., X.H. carried out the historical interpretation of the results; N.R.-R., J.-F.N., M.J.C., O.D. wrote the manuscript. All authors gave final approval for publication and agree to be held accountable for the work performed therein.

Competing interests. At the time of writing, Prof. Matthew Collins was a Board Member of Royal Society Open Science, but had no involvement in the review or assessment of the paper.

Funding. O.D. acknowledges funding from the Fondation Roi Baudouin, Belgium (Fonds Jean-Jacques Comhaire, Grant Agreement No. 2016-P2813310-206264). M.J.C. acknowledges funding from Beasts to Craft (ERC Horizon 2020 Grant Agreement No. 787282) as well as Danish National Research Foundation (DNRF128).

Acknowledgements. The authors acknowledge funding from the Fondation Roi Baudouin, Belgium (Fonds Jean-Jacques Comhaire). They are grateful to the National Library of Luxembourg and the Belgian State Archives for granting access to their collections. O.D. thanks Luke Spindler (BioArCh, University of York) for ZooMS analyses on very first samples. O.D., J.-F.N., N.R.-R. thank J. Vnouček for enlightening discussion on parchment fabrication and visual inspection.

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
