## [Peer Review File · Royal Society Open Science]

Review History

RSOS-210210.R0 (Original submission)

Review form: Reviewer 1 (Brigitte Miriam Bedos-Rezak)

Is the manuscript scientifically sound in its present form?

Yes

Are the interpretations and conclusions justified by the results?

Yes

Is the language acceptable?

Yes

Do you have any ethical concerns with this paper?

No

Have you any concerns about statistical analyses in this paper?

No

Recommendation?

Accept with minor revision (please list in comments)

Comments to the Author(s)

Your pioneering study demonstrates how the collaboration between scientific research and material culture studies enriches the humanities by expanding the informational content of written sources. There is little agreement among conservators about the making of medieval parchment, and the ways in which manufacturing processes may have varied locally. Could a biocodological analysis shed light on these issues? Karl T. Steel is an important scholar of the "Animal Turn," and has much to say about animal husbandry and parchment. What is different about BnL ms. 22? Could the difference be explained in a footnote?

Review form: Reviewer 2**Is the manuscript scientifically sound in its present form?**

Yes

Are the interpretations and conclusions justified by the results?

Yes

Is the language acceptable?

Yes

Do you have any ethical concerns with this paper?

No

Have you any concerns about statistical analyses in this paper?

No

Recommendation?

Accept with minor revision (please list in comments)

Comments to the Author(s)

Does the word 'quire' need clarification? Readers familiar with manuscripts will know, but others interested in the method, but not the material, may not be.

Check usage of the word folio/folios which is not applied consistently. Either use folio/folios (English) or folium/folia (Latin) (likewise, bifolio/bifolios or bifolium/bifolia). Stick with either the Latin or the English usage, but don't use both interchangeably!

Summary.

'all the preserved SINGLE LEAF charters of a single medieval Cistercian monastery'
the genuine production of THE Orval scriptorium'

page 1. the sentence beginning 'charters, ...' needs checking for sense.

Page 1 para 3

Please add a short sentence to explain how these 75 MSS are attributed to Orval. Is this through ex libris inscriptions or medieval catalogues or a combination? This is doubtless explained in the documentation listed in the footnotes, but a sentence added here to clarify the reason for the

attribution to the library would be useful to the reader. Also, if this attribution is made on the basis of one or more medieval catalogues, it would be useful to have an indication of the proportion of the medieval library (as recorded in X year) that is represented by the 75 surviving MSS. It may not be possible to make an accurate statement, but even something as general as 'this represents between 40 and 60% of the books listed in the catalogue for 13XX' would be helpful to the reader

page 2, para 1. a huge VOLUME OF material available for study (??)

direct evidence of monastic secondary production – the meaning of this clause is unclear, rephrase

ditto the last sentence in that paragraph ("–a rare direct evidence ..." is ungrammatical) a rare PIECE OF direct evidence?

Page 3

Line 6: exclusively made of sheep SKIN

page 5

para 2: line 23 cut 'as well as' insert 'as do' (vel sim.) (I would also use 'fewer than' here instead of 'less than')

line 30 – rephrase 'some writings were drawn more carefully ...' – meaning unclear. ?? some texts were copied more carefully ???

line 59: acquisitions MADE in the 13th C

page 6

line 21: S & A use zooarch To conclude that goats were rarely raised in ENGLAND? IN the medieval period (there's a word missing here – about location. England assumed).

Line 30: fewer than 85%. Ditto page 7 line 9 'fewer than 100 folios'

Page 7

Para 3: discussion on quality. A strong correlation between decorated MSS and calf skin might be explained by the superior physical qualities of calf skin as a ground for the application of paint.

Line 59: from the 1240s onwards?? (there's a calfskin charter mentioned above dated to 1237).

Page 8:

Line 39: 'final centuries of the Middle Ages' – an odd expression. 'high middle ages' would be the standard usage

Lines 44-5: rephrase this sentence which makes no sense: Like horn cores and bones, a single skin may have produced multiple folia. (Neither horn cores nor bones can produce multiple folios).

Line 46: is 'dwarfs' the right word? You provide a comparative example in the previous sentence which is numerically greater than the Orval material described in the next sentence. Do you mean 'reverses'? (you go on to say that sheep is more common than calf in the parchment having said that calf is more common than sheep in the bones.

Lines 46-7 sentence beginning Here, unlike ... please provide N# for each element, not absolute numbers for sheep and calf and a % for goats.

Line 53: a reference to the pig numbers is needed

Line 60. Mistake here: bovine and ovine cattle ????

Page 9

Line 7: tendentially – is this the right word? It is used in contexts that are contentious, is that intended here? (and why?).

Line 7: “special preference” rather than “special taste” (I don’t suppose they are licking their books ...)

Line 8: inhabited = wrong word. The special preference for calf parchment demonstrated by the Orval monks

Line 22/23 – here’s the point about calf skin being better for the application of paint. That’s implied here, and could usefully be made explicit.

Line 34 ‘was therefore more apt to be inked and painted’ – awkward phrase. Try, ‘and was therefore a better surface for ink and paint’.

A key point here is the ‘nap’ that can be achieved with calf skin (slightly fuzzy) which absorbs the paint effectively.

Page 10

Para 2: key point. Can you add that this change coincides with the introduction and widespread raising of wool sheep (fatter)?

line 27: a ‘world apart’???

line 51: ‘flagrant’ – an odd word to choose. ‘for the success of sheepskin’ is sufficient.

Decision letter (RSOS-210210.R0)

Dear Professor Deparis

The Editors assigned to your paper RSOS-210210 "A biocodicological analysis of the medieval library and archive from Orval Abbey, Belgium" have now received comments from reviewers and would like you to revise the paper in accordance with the reviewer comments and any comments from the Editors. Please note this decision does not guarantee eventual acceptance.

We invite you to respond to the comments supplied below and revise your manuscript. Below the referees’ and Editors’ comments (where applicable) we provide additional requirements. Final acceptance of your manuscript is dependent on these requirements being met. We provide guidance below to help you prepare your revision.

Please submit your revised manuscript and required files (see below) no later than 21 days from today's (ie 26-Mar-2021) date. Note: the ScholarOne system will ‘lock’ if submission of the revision is attempted 21 or more days after the deadline. If you do not think you will be able to meet this deadline please contact the editorial office immediately.

on behalf of Dr Ed Bolt (Associate Editor) and Kevin Padian (Subject Editor)
openscience@royalsociety.org

Editor Comments to Author:

Thanks for your submission. As you can see, the reviewers have some concerns that need to be addressed in another revision; if you can do so successfully we will accept the paper. However, please address their concerns individually.

In addition, a suggestion was raised about visual accessibility of one of your figures and because this is important to us editorially I ask that you consider the following:

"The supplementary data and diagrams are clearly presented. However, the editor should take a view on the accessibility of the colour scheme for the charts. The team has avoided using red/green, and the colour scheme used is more legible for readers who are 'colourblind'. However, the TONE of the colours for 'sheep' and 'mixed / sheep or goat' is almost identical and would be very hard, probably impossible to distinguish for someone who does not rely on colour for clarity. The best way to test this is to print out the diagrams greyscale. (see?) The colours are fine but the tone could do with greater definition (ie: darker or lighter). This is an editorial decision. The content of the diagrams is good."

Reviewer comments to Author:

Reviewer: 1

Comments to the Author(s)

Your pioneering study demonstrates how the collaboration between scientific research and material culture studies enriches the humanities by expanding the informational content of written sources. There is little agreement among conservators about the making of medieval parchment, and the ways in which manufacturing processes may have varied locally. Could a biocodicological analysis shed light on these issues? Karl T. Steel is an important scholar of the "Animal Turn," and has much to say about animal husbandry and parchment. What is different about BnL ms. 22? Could the difference be explained in a footnote?

Reviewer: 2

Comments to the Author(s)

Does the word 'quire' need clarification? Readers familiar with manuscripts will know, but others interested in the method, but not the material, may not be.

Check usage of the word folio/folios which is not applied consistently. Either use folio/folios (English) or folium/folia (Latin) (likewise, bifolio/bifolios or bifolium/bifolia). Stick with either the Latin or the English usage, but don't use both interchangeably!

Summary.

'all the preserved SINGLE LEAF charters of a single medieval Cistercian monastery'
the genuine production of THE Orval scriptorium'

page 1. the sentence beginning 'charters, ...' needs checking for sense.

Page 1 para 3

Please add a short sentence to explain how these 75 MSS are attributed to Orval. Is this through ex libris inscriptions or medieval catalogues or a combination? This is doubtless explained in the documentation listed in the footnotes, but a sentence added here to clarify the reason for the attribution to the library would be useful to the reader. Also, if this attribution is made on the basis of one or more medieval catalogues, it would be useful to have an indication of the proportion of the medieval library (as recorded in X year) that is represented by the 75 surviving MSS. It may not be possible to make an accurate statement, but even something as general as 'this represents between 40 and 60% of the books listed in the catalogue for 13XX' would be helpful to the reader

page 2, para 1. a huge VOLUME OF material available for study (??)

direct evidence of monastic secondary production - the meaning of this clause is unclear, rephrase

ditto the last sentence in that paragraph ("a rare direct evidence ..." is ungrammatical) a rare PIECE OF direct evidence?

Page 3

Line 6: exclusively made of sheep SKIN

page 5

para 2: line 23 cut 'as well as' insert 'as do' (vel sim.) (I would also use 'fewer than' here instead of 'less than')

line 30 - rephrase 'some writings were drawn more carefully ...' - meaning unclear. ?? some texts were copied more carefully ???

line 59: acquisitions MADE in the 13th C

page 6

line 21: S & A use zooarch To conclude that goats were rarely raised in ENGLAND? IN the medieval period (there's a word missing here - about location. England assumed).

Line 30: fewer than 85%. Ditto page 7 line 9 'fewer than 100 folios'

Page 7

Para 3: discussion on quality. A strong correlation between decorated MSS and calf skin might be explained by the superior physical qualities of calf skin as a ground for the application of paint.

Line 59: from the 1240s onwards?? (there's a calfskin charter mentioned above dated to 1237).

Page 8:

Line 39: 'final centuries of the Middle Ages' - an odd expression. 'high middle ages' would be the standard usage

Lines 44-5: rephrase this sentence which makes no sense: Like horn cores and bones, a single skin may have produced multiple folia. (Neither horn cores nor bones can produce multiple folios).

Line 46: is 'dwarfs' the right word? You provide a comparative example in the previous sentence which is numerically greater than the Orval material described in the next sentence. Do you mean 'reverses'? (you go on to say that sheep is more common than calf in the parchment having said that calf is more common than sheep in the bones.

Lines 46-7 sentence beginning Here, unlike ... please provide N# for each element, not absolute numbers for sheep and calf and a % for goats.

Line 53: a reference to the pig numbers is needed

Line 60. Mistake here: bovine and ovine cattle ????

Page 9

Line 7: tendentially - is this the right word? It is used in contexts that are contentious, is that intended here? (and why?).

Line 7: "special preference" rather than "special taste" (I don't suppose they are licking their books ...)

Line 8: inhabited = wrong word. The special preference for calf parchment demonstrated by the Orval monks

Line 22/23 - here's the point about calf skin being better for the application of paint. That's implied here, and could usefully be made explicit.

Line 34 'was therefore more apt to be inked and painted' - awkward phrase. Try, 'and was therefore a better surface for ink and paint'.

A key point here is the 'nap' that can be achieved with calf skin (slightly fuzzy) which absorbs the paint effectively.

Page 10

Para 2: key point. Can you add that this change coincides with the introduction and widespread raising of wool sheep (fattier)?

line 27: a 'world apart'???

line 51: 'flagrant' - an odd word to choose. 'for the success of sheepskin' is sufficient.

===PREPARING YOUR MANUSCRIPT===

Please ensure that you include an acknowledgements' section before your reference list/bibliography. This should acknowledge anyone who assisted with your work, but does not

qualify as an author per the guidelines at <https://royalsociety.org/journals/ethics-policies/openness/>.

===PREPARING YOUR REVISION IN SCHOLARONE===

- Ensure that your data access statement meets the requirements at <https://royalsociety.org/journals/authors/author-guidelines/#data>. You should ensure that you cite the dataset in your reference list. If you have deposited data etc in the Dryad repository, please include both the 'For publication' link and 'For review' link at this stage.
- If you are requesting an article processing charge waiver, you must select the relevant waiver option (if requesting a discretionary waiver, the form should have been uploaded at Step 3 'File upload' above).
- If you have uploaded ESM files, please ensure you follow the guidance at <https://royalsociety.org/journals/authors/author-guidelines/#supplementary-material> to include a suitable title and informative caption. An example of appropriate titling and captioning may be found at https://figshare.com/articles/Table_S2_from_Is_there_a_trade-off_between_peak_performance_and_performance_breadth_across_temperatures_for_aerobic_sc_ope_in_teleost_fishes_/3843624.

Author's Response to Decision Letter for (RSOS-210210.R0)

See Appendix A.

Decision letter (RSOS-210210.R1)

Dear Professor Deparis,

It is a pleasure to accept your manuscript entitled "A biocodicological analysis of the medieval library and archive from Orval Abbey, Belgium" in its current form for publication in Royal Society Open Science. The comments of the reviewer(s) who reviewed your manuscript are included at the foot of this letter.

Please ensure that you send to the editorial office an editable version of your accepted manuscript, and individual files for each figure and table included in your manuscript. You can send these in a zip folder if more convenient. Failure to provide these files may delay the processing of your proof. You may disregard this request if you have already provided these files to the editorial office. We also need you to send your ESM, please.

on behalf of Dr Ed Bolt (Associate Editor) and Kevin Padian (Subject Editor)
openscience@royalsociety.org

Associate Editor Comments to Author (Dr Ed Bolt):
Associate Editor
Comments to the Author:
Reviewer comments addressed comprehensively. Can now accept.

Reply to editor comments

As suggested by the editor in order to speed up the ms proof preparation, the bibliography format was changed in Vancouver style. DOIs were also added whenever possible.

"The supplementary data and diagrams are clearly presented. However, the editor should take a view on the accessibility of the colour scheme for the charts. The team has avoided using red/green, and the colour scheme used is more legible for readers who are 'colourblind'. However, the TONE of the colours for 'sheep' and 'mixed / sheep or goat' is almost identical and would be very hard, probably impossible to distinguish for someone who does not rely on colour for clarity. The best way to test this is to print out the diagrams greyscale. (see?) The colours are fine but the tone could do with greater definition (ie: darker or lighter). This is an editorial decision. The content of the diagrams is good."

We are attempting to use the same colours for sheep, goat and calf across all studies to aid interpretation. The colours used here for sheep, goat and calf as the same as those used in Fiddymment, S., (...), Collins, M. J. (2015). Animal origin of 13th-century uterine vellum revealed using noninvasive peptide fingerprinting. *Proceedings of the National Academy of Sciences of the United States of America*, 112(49), 15066–15071. The figure was changed to these colours at the request of the PNAS, cf.

<https://www.pnas.org/content/pnas/112/49/15066/F4.large.jpg?width=800&height=600&carousel=1>)

The same colours are used in this manuscript:

- calf #4F81BD
- sheep #DBBD68
- goat #CC6D68

#DBBD68	#D49568	#CC6D68	
The mid-point of these two gives sheep/goat as #D49568. We have used #F3A481 which we believed was more visible, but we can use #D49568 if required.

#F3A481 
In the revised version of this manuscript we have kept the colors of the original submission. Upon editor's request, we can provide new figures with modified color for sheep/goat (#D49568). Color appearance can be checked at <https://www.color-hex.com/color/d49568>.

Reviewer comments

Reviewer: 1

Your pioneering study demonstrates how the collaboration between scientific research and material culture studies enriches the humanities by expanding the informational content of written sources. There is little agreement among conservators about the making of medieval parchment, and the ways in which manufacturing processes may have varied locally. Could a biocodological analysis shed light on these issues? Karl T. Steel is an important scholar of the "Animal Turn," and has much to say about animal husbandry and parchment.

Biocodological analysis allows us to identify parchment species reliably and systematically on a large corpus of medieval manuscripts. Our study demonstrates that we can shed new light on the production of manuscript book and charters. Indeed, choices of species made by the monks and their distribution among quires inform us about how codices were produced. However, species

identification alone cannot give us information about the parchment making itself and to which extent manufacturing processes have varied locally.

What is different about BnL ms. 22? Could the difference be explained in a footnote?

There is nothing especially different about BnL ms. 22. The reason why it is reported to be analysed at MaSUN (Namur) separately from all others is that it was one of the first codex to be sampled and we wanted to double check proteomic results obtained in Namur and York using different instruments. The results obtained at MaSUN were indeed confirmed by BioArch later (samples from BnL ms. 22 were sent to BioArCh after analysis in Namur). For BnL ms. 22, we decided to report on the place where it was analysed first, without mentioning this double check.

Reviewer: 2

Does the word 'quire' need clarification? Readers familiar with manuscripts will know, but others interested in the method, but not the material, may not be.

Done. Briefly defined as "a group of bifolia nested together" according to Muzerelle, *Vocabulaire codicologique* and Gumbert, *Word for codices* (ref. [6] in the ms).

Check usage of the word folio/folios which is not applied consistently. Either use folio/folios (English) of folium/folia (Latin) (likewise, bifolio/bifolios or bifolium/bifolia). Stick with either the Latin or the English usage, but don't use both interchangeably!

Done (Latin form: folium/folia).

Summary.

'all the preserved SINGLE LEAF charters of a single medieval Cistercian monastery'

the genuine production of THE Orval scriptorium'

Done.

page 1. the sentence beginning 'charters, ...' needs checking for sense.

Done. Sentence modified as: "Charters, i.e. medieval deeds on single leaves, are usually dated and thus offer exquisite temporal resolution".

Page 1 para 3

Please add a short sentence to explain how these 75 MSS are attributed to Orval. Is this through ex libris inscriptions or medieval catalogues or a combination? This is doubtless explained in the documentation listed in the footnotes, but a sentence added here to clarify the reason for the attribution to the library would be useful to the reader. Also, if this attribution is made on the basis of one or more medieval catalogues, it would be useful to have an indication of the proportion of the medieval library (as recorded in X year) that is represented by the 75 surviving MSS. It may not be possible to make an accurate statement, but even something as general as 'this represents between 40 and 60% of the books listed in the catalogue for 13XX' would be helpful to the reader

New sentences were added in order to clarify this point: "Several clues make the link obvious or possible between these 118 CUs and the medieval library of Orval. First, 26 CUs were copied in Orval. 33 other CUs are of uncertain origin, but their presence in the medieval library is ascertained by ex-libris dated before the middle of the 16th cent. All the other CUs have a certified provenance at least for the modern period, because they were bound to one of the above-mentioned items, or because their contents are described in a mid-17th-cent. catalogue, or finally because they appear in miscellanies bearing the shelf mark of the second quarter of the 18th cent."

page 2, para 1. a huge VOLUME OF material available for study (??)

Done.

direct evidence of monastic secondary production – the meaning of this clause is unclear, rephrase

Done. We rephrased the sentence: "...they have rarely been considered as primary information on a monastic production, i.e. skins".

ditto the last sentence in that paragraph ("–a rare direct evidence ..." is ungrammatical) a rare PIECE OF direct evidence?

Done.

Page 3

Line 6: exclusively made of sheep SKIN

Done.

page 5

para 2: line 23 cut 'as well as' insert 'as do' (vel sim.) (I would also use 'fewer than' here instead of 'less than')

Done.

line 30 – rephrase 'some writings were drawn more carefully ...' – meaning unclear. ?? some texts were copied more carefully ???

Done: « copied » is OK.

line 59: acquisitions MADE in the 13th C

Done.

page 6

line 21: S & A use zooarch To conclude that goats were rarely raised in ENGLAND? IN the medieval period (there's a word missing here – about location. England assumed).

The geographical indication which opens the sentence ("In England, ...") goes for both Dyer and Salvagno & Albarella. It seems grammatically correct to us, but the editor could change if needed.

Line 30: fewer than 85%. Ditto page 7 line 9 'fewer than 100 folios'

Done.

Page 7

Para 3: discussion on quality. A strong correlation between decorated MSS and calf skin might be explained by the superior physical qualities of calf skin as a ground for the application of paint.

See below, comment on p. 9.

Line 59: from the 1240s onwards?? (there's a calfskin charter mentioned above dated to 1237).

The 1237 charter (made of calf) is the most recent one, and the only one in this decade, which, in our opinion, designates the 1230s (or even the 1220s) as the turning point.

Page 8:

Line 39: 'final centuries of the Middle Ages' – an odd expression. 'high middle ages' would be the standard usage

Done : "in the high and late Middle Ages".

Lines 44-5: rephrase this sentence which makes no sense: Like horn cores and bones, a single skin may have produced multiple folia. (Neither horn cores nor bones can produce multiple folios).

Of course. We changed the sentence in : “Like several horn cores and bones may belong to one single animal, multiple folia may be produced from one single skin.”

Line 46: is ‘dwarfs’ the right word? You provide a comparative example in the previous sentence which is numerically greater than the Orval material described in the next sentence. Do you mean ‘reverses’? (you go on to say that sheep is more common than calf in the parchment having said that calf is more common than sheep in the bones.

Our comparison is based on the study by R. Thomas and colleagues (ref [24] in the ms). Please note this study on bones from domestic livestock in London (including cattle (calf) and sheep, relevant here for comparison with skins, i.e. parchments) spans a vast period, from AD 1220 to 1900. In order to answer accurately to the reviewer’s concern, we made several changes (and corrections) but they did not affect the conclusions reported in the original ms(see below)

First, we corrected previously quoted total numbers of cattle and sheep by looking back to data from Table 3 in ref. [24] (last column): 4300 calf (instead 5084, cattle bones and horn cores counted), 1951 sheep (instead of 2456, ovicrapid bones). Please note these numbers stand for the whole period (1220-1900) of Thomas’ study. We quote these numbers just to fix ideas about the huge volume of bones analyzed in [24]. We also refined our choice of the relevant period (Table 3 in [24]) by selecting the sub-phase A1 (1220-1300) instead of phase A (1220-1350) since this choice “matches” even better the period of interest for comparison with the 20 non-Orval CUs (1st and 2nd third of 13th cent., cf. Fig. of our ms - timeline). For this period (1220-1300), we quote now 129 calf (instead of 287 quoted previously) and 97 sheep (instead of 236 quoted previously). The numbers for the 20 non-Orval CUs had also to be updated in order to represent the actual 158 acquired folia (previously quoted numbers were for all (92) non-Orval CUs).

In spite of these changes in numbers, our original conclusions remain valid. 1) The total number of folia for the 20 non-Orval CUs amounts to 1373 (among which 158 acquired folia) whereas the number of bones (cattle and sheep) is 129+97=226 for closely similar periods; from this point of view, it is legitimate to state that our study (on skins/parchments) “dwarfs” typical zooarchaeological studies (on bones) such as [24]. 2) In our study, sheep is more common than calf (118/29= 4,06).

Numbers were updated accordingly in the revised version of the ms.

Lines 46–7 sentence beginning Here, unlike ... please provide N# for each element, not absolute numbers for sheep and calf and a % for goats.

Numbers were updated (see above).

Line 53: a reference to the pig numbers is needed

Done: reference to R. Noël [26].

Line 60. Mistake here: bovine and ovine cattle ????

Done. Replaced by: “Bovine herds and ovine flocks”.

Page 9

Line 7: tendentially – is this the right word? It is used in contexts that are contentious, is that intended here? (and why?).

Done (word replaced by “usually”).

Line 7: “special preference” rather than “special taste” (I don’t suppose they are licking their books ...)

Done.

Line 8: inhabited = wrong word. The special preference for calf parchment demonstrated by the Orval monks

Done. Sentence modified as "... the special preference for calf parchment apparently demonstrated by...".

Line 22/23 – here's the point about calf skin being better for the application of paint. That's implied here, and could usefully be made explicit.

We fully agree with this hypothesis, which was already developed by Bischoff F. M. 1993 [21], p. 62-71. However, decorated manuscripts are nearly absent from the Orval corpus (two specimens only), which does not allow us to further investigate on this point. The two specimens in question are indeed made of calfskin; yet, they are luxury manuscripts, and therefore various other reasons may account for the selection of calfskin.

Line 34 'was therefore more apt to be inked and painted' – awkward phrase. Try, 'and was therefore a better surface for ink and paint'.

Done.

A key point here is the 'nap' that can be achieved with calf skin (slightly fuzzy) which absorbs the paint effectively.

See our reply above.

Page 10

Para 2: key point. Can you add that this change coincides with the introduction and widespread raising of wool sheep (fatter)?

Done. The following sentence was added: "There might also be a link with the introduction and widespread raising of fatter wool sheep".

line 27: a 'world apart'???

Done: « a reality apart » (change also made in the subsection heading)

line 51: 'flagrant' – an odd word to choose. 'for the success of sheepskin' is sufficient.

Done.